# A Machine Learning Approach to Waterbody Segmentation in Thermal Infrared Imagery in Support of Tactical Wildfire Mapping

**Jacqueline A. Oliver** [1,2,*], **Frédérique C. Pivot** [1], **Qing Tan** [1] , **Alan S. Cantin** [3], **Martin J. Wooster** [4]
and **Joshua M. Johnston** [3]

1   Faculty of Science and Technology, Athabasca University, Athabasca, AB T9S 3A3, Canada;
    fpivot@athabascau.ca (F.C.P.); qingt@athabascau.ca (Q.T.)
2   Northern Forestry Centre, Canadian Forest Service, Natural Resources Canada, 5320-122nd Street,
    Edmonton, AB T6H 3S5, Canada
3   Great Lakes Forestry Centre, Canadian Forest Service, Natural Resources Canada, 1219 Queen St. E.,
    Sault Ste. Marie, ON P6A 2E5, Canada; alan.cantin@nrcan-rncan.gc.ca (A.S.C.);
    joshua.johnston@nrcan-rncan.gc.ca (J.M.J.)
4   Leverhulme Center for Wildfires, Environment and Society, NERC National Centre for Earth Observation,
    Department of Geography, King's College London, London WC2B 4BG, UK; martin.wooster@kcl.ac.uk
*   Correspondence: oliverj21@mytru.ca

**Abstract:** Wildfire research is working toward near real-time tactical wildfire mapping through the application of computer vision techniques to airborne thermal infrared (IR) imagery. One issue hindering automation is the potential for waterbodies to be marked as areas of combustion due to their relative warmth in nighttime thermal imagery. Segmentation and masking of waterbodies could help resolve this issue, but the reliance on data captured exclusively in the thermal IR and the presence of real areas of combustion in some of the images introduces unique challenges. This study explores the use of the random forest (RF) classifier for the segmentation of waterbodies in thermal IR images containing a heterogenous wildfire. Features for classification are generated through the application of contextual and textural filters, as well as normalization techniques. The classifier's outputs are compared against static GIS-based data on waterbody extent as well as the outputs of two unsupervised segmentation techniques, based on entropy and variance, respectively. Our results show that the RF classifier achieves very high balanced accuracy (>98.6%) for thermal imagery with and without wildfire pixels, with an overall F1 score of 0.98. The RF method surpassed the accuracy of all others tested, even with heterogenous training sets as small as 20 images. In addition to assisting automation of wildfire mapping, the efficiency and accuracy of this approach to segmentation can facilitate the creation of larger training data sets, which are necessary for invoking more complex deep learning approaches.

**Keywords:** waterbody segmentation; wildland fire; thermal infrared; random forest

## 1. Introduction

Wildfires are common around the globe, consuming an average of ~422.5 million hectares (Mha) annually [1]. In Canada, wildfires burn ~2 Mha of forest each year, with a few large fires (>200 ha) typically considered responsible for 97% of the annual area burned [2,3]. Over the last several decades, the frequency of these large wildfires has been increasing in Canada, along with the annual area burned and fire season length [3]. With changing climate conditions, these trends are expected to intensify [4–7]. While it is increasingly common to allow fires considered low-risk to burn as a natural component of healthy ecosystems, fire suppression activities are necessary, notably when there is a risk to human lives, infrastructure, or other values [8]. Increasing fire activity due to climate

change, coupled with expanding areas at risk due to growth and reshaping of wildland–urban interface (WUI) areas [9], amounts to intensified pressure on fire management agencies. The result is escalating monitoring and suppression costs that have typically not been matched with program budget increases [8]. In response, many agencies are exploring the potential for emerging technologies to provide efficient and cost-effective tools to assist decision making and facilitate safe, effective, and rapid response to wildfire activity.

Airborne thermal infrared (IR) imagery can aid wildfire response by offering very fine spatial resolution imagery (e.g., <5 m) that renders smoke largely invisible and makes fire easy to distinguish from surrounding ground cover [10]. In the 1980s, forward-looking infrared (FLIR) imagery was used to improve flame retardant delivery and application, enabling air attack supervisors to view an image of the fire through the smoke to better determine areas of elevated fire intensity [11]. More recently, nighttime airborne overflights have been used to produce fine spatial resolution heat mapping of hazardous wildfires to aid fire response operations [10,12]. Although the technology to collect such data has become increasingly available and affordable, the application of techniques to extract information rapidly and accurately from the collected airborne IR imagery has not proceeded at the same pace. This is due to both lack of radiometric calibration in some airborne IR imagery and the operational domain remaining reliant on manual processes for isolating fire pixels (both smouldering and flaming) within the imagery. As such, the process is time-consuming and dependent on human interpreters and their expert judgment [13]. Reliance on human expert intervention is not just slow; it results in inconsistent outputs among interpreters and renders real-time mapping implementations unworkable.

In recent years, the wildfire research community has taken significant steps toward the automation of tactical wildfire mapping products (e.g., [14]). Computer vision (CV) techniques have been successfully applied to aerial image processing in the context of wildfire management. Valero et al. [15] proposed a combination of thresholding, morphological processing, and active contouring to map fire perimeters based on thermal IR imaging. Flame height, front, and depth have also been explored in both visible and IR imagery using histogram thresholding and heuristics; such thresholding often results in noise or false positives, so it is typically followed by contouring or noise corrections [16]. Many of the applied techniques for segmenting areas of interest in remote sensing imagery use some form of edge detection [14,17] or rely on colour rules made possible by the inclusion of visible spectrum images [18].

In Canada, the Canadian Forest Service (CFS) has implemented Torchlight, a national tactical wildfire mapping system. Torchlight leverages contextual image processing and CV techniques to produce detailed fire perimeters for all available data and can discriminate between areas of intense, scattered, isolated, or residual heat [19]. This system has been successfully deployed during wildfire emergencies that threatened lives or significant material and cultural values [20]. Nevertheless, irregularities in the source imagery caused by the relative overnight warmth of lakes and/or reflections from the sun or moon from their surface (at shorter IR wavelengths) [21] result in some waterbodies being incorrectly marked as wildfire during image processing operations. For this reason, systems such as Torchlight still rely on human intervention to manage the false positive identification of waterbodies as fire pixels. A simple static water mask is often not sufficient to remove these false positives because areas of smaller waterbodies can change quite significantly throughout a season.

Studies assessing the segmentation of waterbodies in aerial imagery are numerous but, to date, have focused primarily on using visible light or multispectral imagery [22–24]. Single-band threshold value techniques have been used extensively to isolate water pixels, but success hinges on spectral characteristics found at visible and near-infrared wavelengths [25,26]. Other popular solutions include band-ratio methods and those based on the spectral water index, such as normalized difference water index (NDWI). These techniques exploit differences in the spectral reflectance of surfaces at different wavelengths, particularly between the green and shortwave infrared bands [27]. While established as

effective, this dependence renders them useless for a single-dimension domain (i.e., one waveband). In the context of wildfire mapping and monitoring, the inclusion of or reliance on visible imagery is also impractical as it cannot be gathered overnight when the thermal contrast between ambient ground cover and fire is at its peak (particularly for low-intensity smouldering combustion that could be the source of intense fire activity in the next burning cycle). Moreover, during the day, the visible channel data of a fire may be heavily occluded by smoke.

More promising solutions for this problem are found in studies that have focused on incorporating texture into classification or segmentation techniques. Nath and Deb [24] tested entropy-based waterbody extraction on optical and synthetic aperture radar (SAR), concluding that it was most successful for optical images but needed to be combined with post-processing to smooth the output. Rankin and Matthies [28] also used texture as an indicator of water by considering variance in $5 \times 5$ pixel regions of an image. However, their study was based on colour imagery, with the greyscale variance being used to identify candidate water regions as a precursor to examining colour changes.

Texture has also been considered for other aspects of wildfire mapping. Hamilton et al. [29] evaluated adding texture as a fourth input for high-resolution wildfire severity mapping in visible light imagery using a support vector machine (SVM). Their texture metrics were based on those proposed by Haralick [30], including first-order calculations based on pixel neighbourhood and second-order calculations based on the grey-level co-occurrence matrix (GLCM). They concluded that there were statistically significant gains in accuracy from all texture parameters considered, with each input in isolation contributing nearly as much information gain as the green or blue light band [29]. Due to concerns about computational complexity, their study did not investigate whether the inclusion of multiple texture parameters could further improve accuracy. Smith et al. [31] also considered GLCM-based textures, examining their application for burn scar detection based on a single band. They concluded that texture could highlight burned areas but ultimately that it did not improve accuracy in their single-band application.

A segmentation algorithm based exclusively on thermal IR data presents unique challenges. The image produced depends on the object's apparent temperature (brightness), which is impacted, for example, by its surface properties (most notably thermal IR emissivity) and its orientation with respect to the sensing device. This translates to images that typically have reduced contrast and spatial precision [32]. As a result, these thermal images often cannot be reliably processed using the same techniques or features that are successful for the segmentation of visible spectrum imagery. Furthermore, while some of the established techniques may be successfully applied to imagery collected exclusively in the thermal IR spectral band, the ability to assess the utility of the output has been limited by the size and extent of training and validation data available to implement and evaluate them [33]. To date, no studies have assessed the effectiveness of existing waterbody segmentation techniques in thermal IR imagery collected during wildfires, despite reports that the presence of fire-impacted areas in imagery may reduce their efficacy [11]. In response, the aim of this study was to:

(i) Evaluate various methods for waterbody segmentation in thermal IR imagery collected over areas of varying terrain with and without the presence of flaming and smouldering combustion;

(ii) Compare segmentation method outputs and existing publicly available static GIS waterbody boundary layers against reference data created from the aerial images and discuss the implications.

This study considers several methods for waterbody segmentation in IR images, including unsupervised techniques based on variance and entropy, as well as supervised classifiers using random forest. Random forest (RF) is a classic machine learning algorithm that uses bagging to construct a set of decision trees in which each tree is based on a subset of the input data features [34]. Establishing an RF classifier requires labelled data (i.e., reference data), both for training the classifier itself and for validating the classifier's

performance after training. Fortunately, an RF classifier can be successful with significantly less labelled data than would be required for deep learning algorithms. This study explores the use of RF classification to segment thermal IR imagery based on the spectral data, as well as texturally derived features to generate contextual awareness in the RF. The results of each image segmentation method are compared against the results achieved by relying on publicly available static GIS data.

## 2. Materials and Methods

### 2.1. Study Area

The study area is located within Woodland Caribou Provincial Park, west of the municipality of Red Lake, Ontario, Canada (Figure 1). In this remote area, the rolling and rugged Canadian shield rock overlaps the boreal forest. It boasts an abundance of lakes, rivers, and wetlands with footprints that shift with the area's high levels of precipitation, long cold winters, and short warm summers.

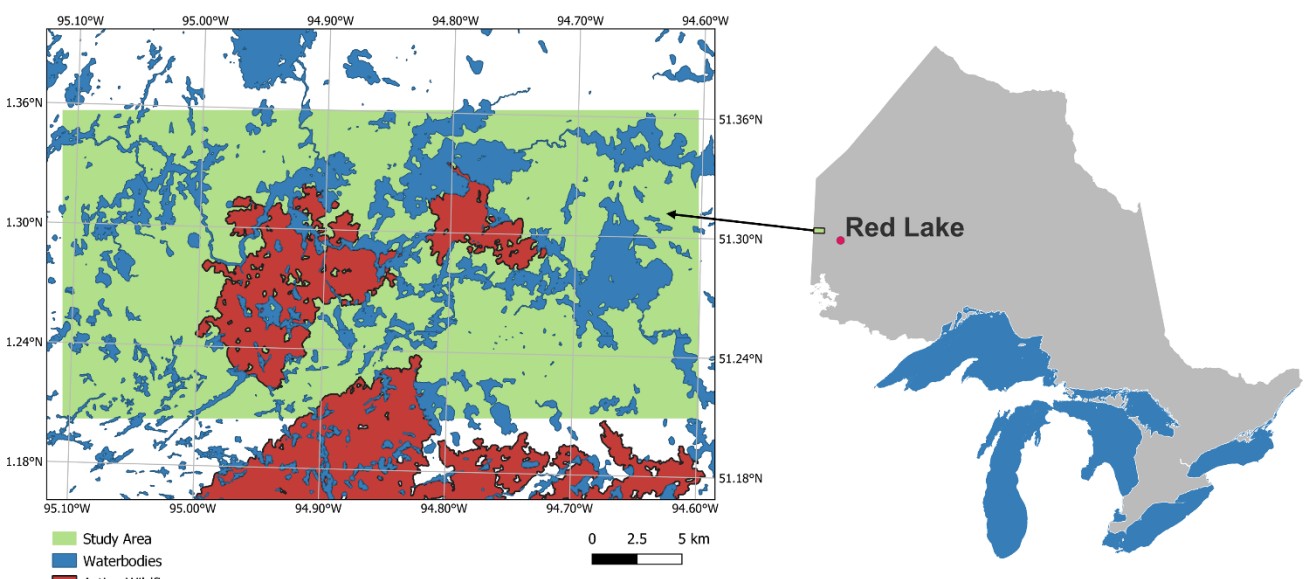

**Figure 1.** Thermal imagery study area in Woodland Caribou Provincial Park, near Red Lake Ontario. Map displays the wildfires that were active at the time of image collection, on 12 August 2018, based on the National Burned Area Composite (NBAC) of Canada [35].

### 2.2. Methodology

#### 2.2.1. Airborne Imagery

This study uses a subset of 2496 geocoded and orthorectified (with ~2 m geolocation accuracy) airborne thermal images collected in the mid-wave infrared (MWIR) over an area of active large wildfires (Figure 1). The overnight flight was conducted on 12 August 2018; data collection began at 23:00 and was completed by approximately 23:30. The data were collected using a nadir viewing thermal imager fitted onboard a DHC-6 Twin Otter equipped with a FLIR SC6703 thermal imaging camera flying at approximately 8000–10,000 feet above ground level, depending on cloud conditions present at the time. The camera used employs an indium antimonide (InSb) detector to provide imagery in the MWIR thermal band (camera spectral band is 1.0–5.0 μm but is limited to ~3.85–3.95 μm with the aid of a spectral filter) at a nominal frame rate of typically 30 Hz. The imagery spatial resolution from the DHC-6 flying height was approximately 1 m, and images were collected during each sortie above the target area sampled using sequential overlapping ~10 km transects that covered multiple highly heterogeneous wildfires during the day and

overnight. The resulting data were subset to those containing features of interest for this study. For this study, images were processed and analyzed individually.

### 2.2.2. Ancillary Data

The CanVec hydrographic features data set [36] was used to provide a baseline representation of the accuracy attainable from existing static waterbody representations. This CanVec data series contains vectorized geospatial representations of water features across Canada. The finest scale version was used, which shows waterbodies at a scale of 1:50,000.

### 2.2.3. Training and Validation Reference Data

A reference data set that could later be divided into training and validation data was created. The IR images in the study data set were sequential, with a heavy overlap between frames. After discarding blank and low-quality images, images were selected at pseudorandom intervals to ensure a variety of terrain was captured while reducing replication in the training/validation data set. In addition, targeted selections were added to ensure coverage of heterogenous wildfire patterns. This was necessary, as less than one-third of all images contained wildfire pixels and those that did tend to appear in clusters. Among fire-containing images, there was significant variation in the intensity and extent of wildfire pixels. This attests to the mixed flaming and smouldering nature of fires during nighttime conditions. Ultimately, the test data set displayed dramatic variation in waterbody density (~1–98% of the image area) and wildfire density (~0–15% of the image area).

Targeted image selection was performed by comparing the ratio of maximum pixel brightness temperature to minimum non-zero temperature in each image. Images with ratios that exceeded 1.2 (the threshold associated with no fire activity) were sorted in increasing order. Images were selected at pseudorandom intervals to ensure that sufficient time had passed in the flight transect to provide distinctive scene structures (~8–12 frame intervals depending on airspeed). Subsequently, additional frame selections were made among lower ratio images to reflect their elevated frequencies in the overall data set. This resulted in 18 additional images being selected, for a total of 267 images.

Due to inaccuracies and incompleteness observed in existing spatial waterbody data sets, as well as changes in water levels and seasonal inundation of wetlands, neither existing geospatial vector data nor high-resolution static reference imagery were used to determine the actual boundaries of waterbodies. Instead, high-quality training data were created through manual delineation of the IR images following standard protocols applied in operational airborne wildfire mapping applications. In this process, all visible non-transient waterbodies (e.g., including only semi-permanent bodies such as lakes, rivers, creeks, etc.) were identified. No minimum waterbody size requirements were applied. Delineation was performed by a single observer (J. Oliver) to ensure consistency in interpretation and to avoid interobserver bias.

For the images selected to be in the reference data set, delineation of waterbody boundaries was completed through the creation of polygons in QGIS [37]. To improve contrast and visibility for visual inspection, the no-data value of 0 was set to display as transparent, while extreme intensities from fire pixels were capped at lower values (Figure 2A). For each thermal IR image, a polygon was created for each waterbody by tracing its perimeter and adjusting vertices incrementally (Figure 2B). Waterbodies are typically texturally distinct from surrounding terrain, so the manual determination of their borders was a reliable albeit time-consuming process. When necessary, confirmation of the general location of waterbody boundaries was determined by comparing the IR imagery against existing visible satellite imagery and the CanVec series hydrographic features vector data [36]. Where edges or boundaries were ambiguous, the edge of the waterbody was judged by textural consistency.

The final reference images based on the delineated waterbody shapes were created using a rasterize function [38] in Python (Figure 2C), with each cell overlapping a waterbody set to the value one.

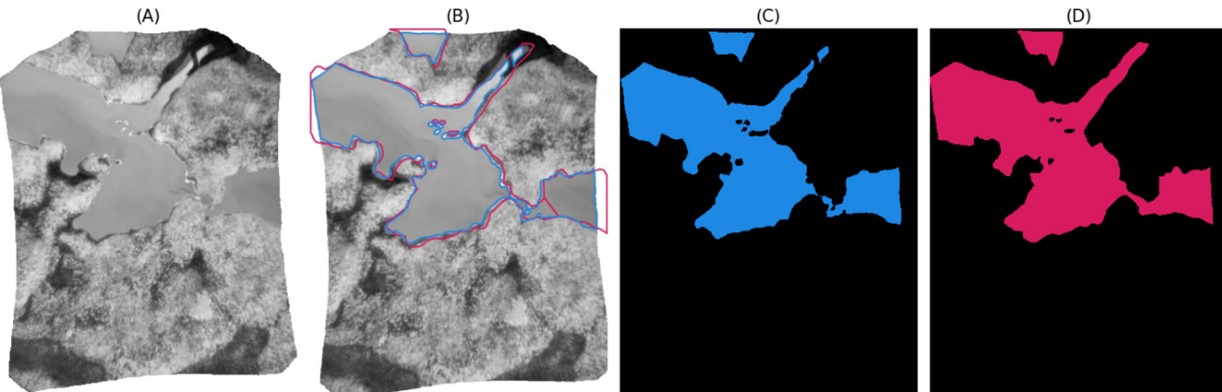

**Figure 2.** (**A**) Original raster image, with no-data values masked out for figure visibility; (**B**) original Raster with delineated waterbody boundary (blue) and existing CanVec boundary (pink) overlaid; (**C**) reference raster generated based on original raster and waterbody boundary; (**D**) Canvec Raster generated based on original raster image, adjusted to reflect no-data border in the original.

Comparison images demonstrating the accuracy attainable by relying on static waterbody GIS data were created by rasterizing the CanVec hydrographic feature set [36] for each reference image (Figure 2A,B). To account for the no-data border that surrounds each IR image, any pixel that was set to no-data in the original image was set to no-data in the CanVec comparison raster (Figure 2D).

Twenty of the prepared reference images were randomly selected as an efficient base data set to identify potentially useful features and then assess their importance for RF classification. This set included 10 randomly selected fire pixel-containing photos and 10 randomly selected photos distributed across the terrain. For the final evaluation, the RF classifier was trained on randomly selected image sets of 20, 35, and 50 images, which included 10, 12, and 15 fire pixel-containing reference images of varying intensity, respectively. The trained models were then validated by classifying all remaining images that had an associated reference image created.

### 2.3. Classifier Feature Identification

Normalization processes, existing image filters, and unique image filter combinations were considered as potential inputs for the RF classifier. The full details of these features are discussed in Appendix A.

Data normalization was necessary to address the anticipated loss of detail from passing pixels with very small values to standard image processing libraries, which often truncate long input values. Although there is significant utility in the pixel measurements, in an operational setting, it is uncommon for the input data to be radiometrically calibrated. For this reason, original pixel values (i.e., brightness temperature) were retained as input features for the model but were generally not used as inputs to image filters. Instead, normalized versions that rescaled the pixel values to fall between 0 and 1 were used on a per-image basis. A second form of normalization (max-normalization hereafter) was also tested to account for the skewing caused by the relatively large values found exclusively in images with fire activity. This technique selects a maximum pixel value as some multiple of the minimum non-zero pixel value. Any pixel value that exceeds this maximum is set to be equal to it prior to normalizing the values to range between 0 and 1.

Existing rank image filters were the primary source of feature generation for the RF classifier. These filters use the local neighbourhood of a pixel (i.e., surrounding pixel values) to assign a new value to each pixel, enabling them to capture the context of isolated pixels. The image filters considered are listed in Table 1. These features were generated using standard Python image processing library functions [39,40]. Secondary filters were applied to the outputs of these filters to generate further variations (Table 1). In addition, six texture

measurements introduced by Haralick et al. [30] were included as features. Each of these is derived from a grey-level co-occurrence matrix (GLCM).

**Table 1.** Image filters and variants tested for RF feature generation.

| Filter Name | Description/Equation | Filter Input | | | Kernel Shape | | Kernel Sizes [a] | Secondary Filter Applied [b] | | | |
|---|---|---|---|---|---|---|---|---|---|---|---|
| | | Orig. | Norm | Max-Norm | Disk | Square | | Min | Max | Var | Ent |
| **Standard Image Filters** | | | | | | | | | | | |
| Entropy (Ent) | Local entropy using base 2 log. | | ✓ | ✓ | ✓ | ✓ | 2–5, 7, 15, 25, 55 | ✓ | ✓ | ✓ | ✓ |
| Mean | Local mean value. | | ✓ | ✓ | ✓ | ✓ | 2–5, 7, 15, 25, 55 | ✓ | ✓ | ✓ | ✓ |
| Subtracted Mean | Difference between centre value and local mean value. | | ✓ | ✓ | ✓ | ✓ | 2–5, 7, 15, 25, 55 | ✓ | ✓ | ✓ | ✓ |
| Sum | Sum of local values. | | ✓ | ✓ | ✓ | ✓ | 2–5, 7, 15, 25, 55 | ✓ | ✓ | ✓ | ✓ |
| Threshold | Local threshold value. | | ✓ | ✓ | ✓ | ✓ | 2–5, 7, 15, 25, 55 | ✓ | ✓ | ✓ | ✓ |
| Variance (Var) | Average of squared differences from mean. | | ✓ [c] | ✓ [c] | | ✓ | 2–10 | ✓ | ✓ | ✓ | ✓ |
| **Grey-Level Co-Occurrence Matrix (GLCM) Filters [d]** | | | | | | | | | | | |
| Angular Second Moment (ASM) | $\sum_{i,j=0}^{N-1} P_{i,j}^2$ | | ✓ | ✓ | | ✓ | 7 | | | | |
| Contrast | $\sum_{i,j=0}^{N-1} P_{i,j}(i-j)^2$ | | ✓ | ✓ | | ✓ | 7 | | | | |
| Correlation | $\sum_{i,j=0}^{N-1} P_{i,j}(i-j)^2$ | | ✓ | ✓ | | ✓ | 7 | | | | |
| Dissimilarity | $\sum_{i,j=0}^{N-1} P_{i,j}\left[\dfrac{(i-\mu_i)(j-\mu_i)}{\sqrt{(\sigma_i^2)(\sigma_j^2)}}\right]$ | | ✓ | ✓ | | ✓ | 7 | | | | |
| Energy | $\sqrt{ASM}$ | | ✓ | ✓ | | ✓ | 7 | | | | |
| Homogeneity | $\sum_{i,j=0}^{N-1} \dfrac{P_{i,j}}{1+(i-j)^2}$ | | ✓ | ✓ | | ✓ | 7 | | | | |
| **Combination Image Filters** | | | | | | | | | | | |
| Scaled Entropy Stack (SE$_S$) | Local entropy calculated with increasing capped max pixel values. Matrices merged, storing max entropy calculated at each pixel. | ✓ | ✓ | ✓ | ✓ | ✓ | 2–5, 7, 15, 25, 55 | ✓ | ✓ | ✓ | ✓ |
| Minimum Shifted Entropy (SE$_{min}$) | Local entropy for pixel at centre, top, bottom, far left, and far right of kernel. Minimum value stored. | | ✓ | ✓ | ✓ | | 2–5, 7, 15, 25, 55 | ✓ | ✓ | ✓ | ✓ |
| Maximum Shifted Entropy (SE$_{max}$) | Local entropy for pixel at centre, top, bottom, far left, and far right of kernel. Maximum value stored. | | ✓ | ✓ | ✓ | | 2–5, 7, 15, 25, 55 | ✓ | ✓ | ✓ | ✓ |
| Binary Variance (BV) | Local zero-variance pixels with are grown into regions through morphological dilation and erosion. | | ✓ [c] | ✓ [c] | | ✓ | 2, 3, 4 | | | | |
| Binary Entropy (BE) | Local low entropy pixels are selected as water. Minimum filter passes reduce noise, followed by a maximum filter to regrow lost area. | | ✓ [c] | ✓ [c] | ✓ | ✓ | 2–5, 7, 15, 25, 55 | | | | |

[a] Kernel size refers to pixel radius of a disk-shaped kernel and pixel width of square-shaped kernel. [b] Secondary filters were applied with disk-shaped kernels at sizes of 3, 5, 7. [c] Input was median filtered prior to processing. Median filters were applied at square kernel sizes of 2, 3, 4, 5, 7, and 15. [d] These metrics are based on those defined by Haralick [30]. Formulas are from the implementations presented in the Scikit package [39]. $p$ is the GLCM histogram based on the region of the image, $N$ is the size of the GLCM, $\mu$ is GLCM mean, and $\sigma^2$ is variance.

Lastly, five unique filter combinations were created as feature inputs (Table 1). The scaled entropy stack (SE$_S$) uses iterative max-normalization in conjunction with entropy calculations to retain the maximum possible entropy value at each pixel. Minimum shifted entropy (SE$_{min}$) and maximum shifted entropy (SE$_{max}$) calculate local entropy from five different positions within a local neighbourhood and retain only the minimum or maximum value, respectively. Binary variance (BV) was inspired by the rule-based approach to waterbody segmentation that was introduced by Wilson [41]. This feature calculates variance using a 2 × 2 sliding window, retains only pixels that have zero variance, then expands these zero-variance regions using several iterations of morphological dilation and erosion. The binary entropy (BE) feature follows a similar procedure to BV but is based on a minimal entropy threshold rather than variance. The BV and BE techniques developed here are also tested as unsupervised waterbody segmentation methods. A fuller explanation of their implementation is provided in Appendix A.

### 2.4. Waterbody Segmentation Methods

Five different approaches for automating the detection and segmentation of waterbodies in the thermal IR imagery were considered: static GIS data, binary entropy (unsupervised), binary variance (unsupervised), a random forest classifier trained only on individual pixel values, and a random forest classifier trained using context and texture features.

### 2.4.1. Static GIS Data Segmentation

To generate a baseline comparison, the CanVec comparison images were analyzed against the reference rasters that were generated from the IR images. This was carried out by comparing the individual values for each pixel and each position between the CanVec generated image and the reference raster.

### 2.4.2. Unsupervised Techniques: Binary Entropy and Binary Variance Filter Combinations

The binary entropy and binary variance filter combination were run for each image that had an associated reference raster. Each of these methods runs its respective sequence of filters described in Table 1, producing a binary output matrix. These direct outputs were considered through pixel-by-pixel comparisons against the reference data. No post-processing methods were applied.

### 2.4.3. Baseline Random Forest Classifier

Three versions of an RF classifier were trained to assess classifier effectiveness based solely on isolated pixel values. The first version used only original pixel values. The second was trained using both the original pixel values and the pixel values normalized to a range between 0 and 1. The final version was trained using original pixel values, normalized pixel values, and the max-normalized version that set any value greater than 1.2 times the minimum non-zero ($Val_{min}$) to be equal to it prior to normalizing between 0 and 1. Training was conducted on all three training data sets (20, 35, and 50 images) and on the no-fire data set.

### 2.4.4. Random Forest Classifier with Feature Selection

The features identified as potential inputs for the RF classifier were generated using a variety of kernel sizes and two unique kernel shapes (where alternate kernel shapes were supported), as detailed in Table 1. Due to the extremely large total number of feature candidates this produced, it was essential to assess features individually for their ability to contribute information gain to classification prior to testing feature importance as a group. Each feature output from Table 1 was tested by training a 100-tree RF classifier with that feature as well as the original pixel (i.e., brightness temperature) value and normalized values. This training was performed using 75% of the image data from the subset of 20 images, with 25% of the data reserved for calculating performance metrics. The F1 score for each model was calculated based on this reserved data. For variance- and entropy-based features, any output that could produce a classifier with an F1 score greater than 0.90 was retained for further consideration. This very high threshold standard was selected due to the strong performance attained by these methods (typically well above 0.70). For all other features, the cut-off threshold was an F1 score > 0.70.

All feature variants that exceeded the necessary F1 scores were used together as feature inputs to train an RF classifier of 100 trees with no maximum depth. Each image was loaded and passed through the identified filters to produce variations. The resulting images were reshaped into one-dimensional columns that, when combined, produced a matrix where each row represents the various values for a single pixel. Training was based on resulting matrices from the subset of 20 images, with 75% of the image data being used for training the classifier and 25% reserved for validating the model.

Once trained, feature importance was assessed using Gini impurity, as implemented in the Scikit-learn package [42]. Any feature that had an importance of 0 (i.e., no impact on classification) was removed. The features of lowest importance were then also removed;

these were initially removed in groups of 50, when features tended to have uniformly negligible importance, and in groups of 15 once the number of remaining features was below 200 and their impacts or importance were more pronounced. After feature removal, the classifier was retrained on all remaining features. This process of training and eliminating the least significant features was repeated until elimination resulted in significant drops in accuracy.

Due to the high occurrence of seemingly similar features in the final feature list, a secondary smaller feature list was produced based on feature correlation. The goal was to assess whether it was possible to make a more efficient classifier without a significant loss of accuracy. Similar features (e.g., multiple versions of the same image filter using similar kernel sizes) were reduced by considering pairwise feature correlation and eliminating variables with a Pearson's correlation coefficient > 0.9. This analysis was performed in R using the *cor* and *findCorrelation* functions in the caret package [43]. No-data pixels were removed prior to analyzing correlation, and a minimum of one of each image filter type was retained.

Once optimal feature lists were determined, the optimal number of trees was assessed by training the classifier with increasing numbers of trees. Tree count started at 10 and was increased by 10 at each iteration until 500 trees was reached. No maximum tree depth was assigned.

*2.5. Waterbody Segmentation Method Evaluation*

The outputs from each segmentation technique were compared pixel-to-pixel with the reference images. Based on direct pixel comparison, the accuracy, balanced accuracy, F1 score, precision, and recall scores (Table 2) were calculated using all pixels for each individual image. Direct pixel comparison was also used to calculate these metrics across the entire validation set by summing the total pixel counts (i.e., false negatives, false positives, true negatives, and true positives) present in all validation images. In addition, the runtime to produce the segmentation output was recorded for each individual image as well as generalized across the entire data set. Image read and write times were not considered in this analysis.

**Table 2.** Summary of performance equations.

| Metric | Equation |
|---|---|
| Accuracy | $\dfrac{(TN + TP)}{TP + TN + FP + FN}$ |
| Balanced Accuracy | $\dfrac{\left(\frac{TP}{TP + FN}\right)+\left(\frac{TN}{TN + FP}\right)}{2}$ |
| F1 Score | $\dfrac{2*(Precision*Recall)}{Precision + Recall}$ |
| Precision | $\dfrac{TP}{TP + FP}$ |
| Recall | $\dfrac{TP}{TP + FN}$ |

Parameters: *TN*, true negatives (i.e., number of pixels correctly classified as non-water); *TP*, true positives (i.e., number of pixels correctly classified as water); *FN*, false negatives (i.e., number of pixels incorrectly classified as non-water); *FP*, false positives (i.e., number of pixels incorrectly classified as water).

## 3. Results

*3.1. Baseline Static GIS Metrics Results*

The CanVec data set produced an average overall balanced accuracy of 0.965, with a minimum image balanced accuracy of 0.752 and a maximum balanced accuracy of 0.993. Overall precision score (i.e., percentage of pixels classified as water that are accurately classified) was 0.897, with individual image minimum and maximums ranging from 0.276 to 0.977. Overall recall score (i.e., percentage of water pixels that were classified as water) was 0.949, with individual image scores ranging from 0.507 to 0.999. Recall and precision scores were poorest when the image had minimal water (i.e., <1%). Most incorrect pixels for CanVec images were due to misplaced waterbody boundaries, which often show a strong directional skew. The results are summarized in Figure 3 and Table 3.

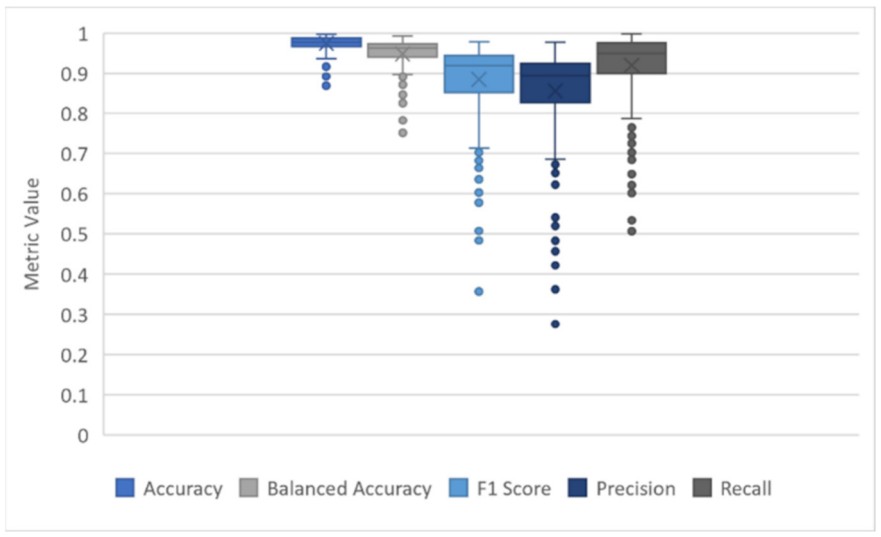

**Figure 3.** Metrics for CanVec baseline data.

**Table 3.** Evaluation metrics for waterbody segmentation methods. Standard deviations presented in brackets. Optimal values for each metric presented in bold.

| | Static GIS Data (CanVec) | Binary Entropy | Binary Variance | Random Forest Classifier | | | | | |
|---|---|---|---|---|---|---|---|---|---|
| **Number of Features** | 1 | 1 | 1 | 21 | 21 | 21 | 91 | 91 | 91 |
| **Number of Training Images** | - | - | - | 20 | 35 | 50 | 20 | 35 | 50 |
| **Accuracy** | 0.975 (0.017) | 0.975 (0.012) | 0.976 (0.013) | 0.988 (0.009) | 0.992 (0.005) | 0.992 (0.005) | 0.989 (0.008) | 0.992 (0.004) | **0.993** (0.005) |
| **Balanced Accuracy** | 0.965 (0.041) | 0.953 (0.050) | 0.947 (0.062) | 0.971 (0.052) | 0.980 (0.042) | 0.981 (0.044) | 0.977 (0.042) | 0.983 (0.037) | **0.984** (0.039) |
| **F1 Score** | 0.923 (0.094) | 0.921 (0.115) | 0.921 (0.119) | 0.961 (0.081) | 0.973 (0.074) | 0.974 (0.078) | 0.966 (0.071) | **0.976** (0.064) | **0.976** (0.067) |
| **Precision** | 0.897 (0.109) | 0.923 (0.132) | 0.938 (0.115) | 0.976 (0.058) | **0.983** (0.053) | 0.982 (0.063) | 0.973 (0.067) | **0.983** (0.050) | 0.982 (0.055) |
| **Recall** | 0.949 (0.083) | 0.920 (0.102) | 0.905 (0.128) | 0.945 (0.104) | 0.963 (0.086) | 0.966 (0.088) | 0.960 (0.083) | 0.969 (0.075) | **0.971** (0.078) |
| **Average Feature Generation Time (s)** [a] | 19.620 [b] | **0.519** | 1.368 | 10.268 | 10.268 | 10.268 | 34.693 | 34.693 | 34.693 |
| **Average Classification Time (s)** [c] | - | - | - | **0.349** | **0.349** | **0.349** | 0.446 | 0.446 | 0.446 |
| **Average Total Processing Time per Image (s)** | 19.620 | **0.519** | 1.368 | 10.617 | 10.617 | 10.617 | 35.139 | 35.139 | 35.139 |

[a] This study used a modified Dell Precision 7550 laptop to compute all runtimes. This system uses an Intel Core i9-10885H CPU at 2.40 GHz with 8 cores and 16 logical processors. There is 64.0 GB (63.69 GB usable) installed RAM. Program files and data were stored locally on a PC611 NVMe SK hynix 1 TB solid-state drive. [b] Time taken to generate mask of existing shapefile data to input IR image; expense is heavily impacted by input file size. [c] Classification time is based on using 70 trees; the optimal number of trees for this data set is based on testing performance for 10–500 trees.

### 3.2. Unsupervised Technique Results

The binary entropy (BE) technique was by far the fastest segmentation method tested. Running at an average of 0.519 s per image, it took less than half the processing time required by the next most efficient method. It produced an overall balanced accuracy of 0.953 with a minimum image balanced accuracy of 0.564 and a maximum balanced accuracy of 0.984. The overall precision score was 0.923, with individual image minimum and maximums ranging from 0.330 to 0.982. The overall recall score was 0.920, with individual minimum and maximums ranging from 0.128 to 0.979. Very poor recall performance (<0.50) occurred exclusively in IR images that had minimal water in the frame; these images had less than 0.6% of the image composed of water pixels, whereas the average image in this set was composed of 16.38% water pixels. Poor precision performance (<0.50) was seen in images with a low proportion of water as well as in images that contained narrow

waterbodies. In images that contain fire, the performance metrics each dropped an average of ~2–4%. The results are summarized in Figure 4A and Table 3.

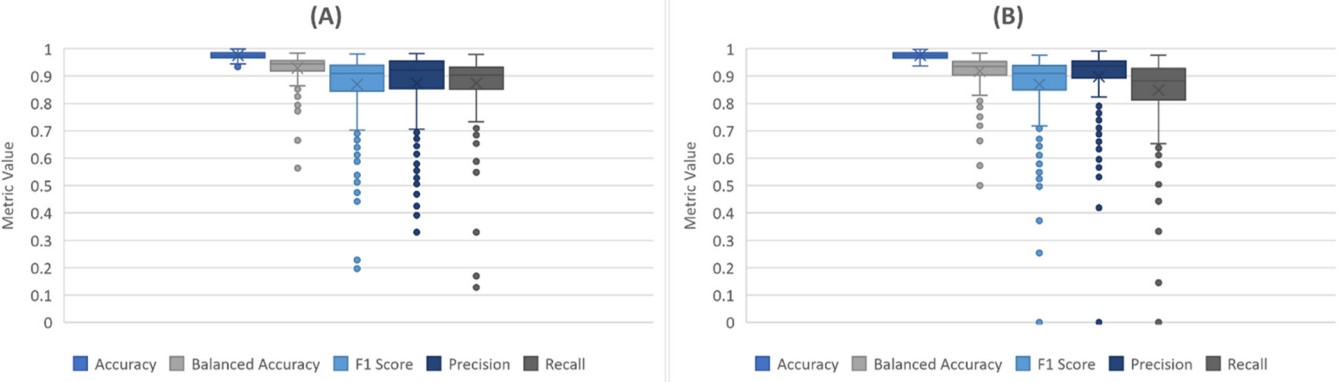

**Figure 4.** (**A**) Metrics for binary entropy unsupervised technique; (**B**) metrics for binary variance unsupervised technique.

The binary variance (BV) technique was slightly less accurate than BE and required nearly double the processing time. It attained an overall balanced accuracy of 0.9468, with individual image values ranging from 0.500 to 0.984. In two images, the technique failed to detect a single correct water pixel; this yielded the minimum F1, precision, and recall scores of zero. As with the BE technique, BV was least successful in images with a low proportion of water pixels and in images with narrow waterbodies. Notably, while images containing fire still showed an overall reduction in performance metrics, there was no impact on precision. The results are summarized in Figure 4B and Table 3.

### 3.3. Spectral Random Forest Classifier Results

When classifying on input values alone (i.e., brightness temperature), the baseline RF classifier was not able to correctly classify even a single water pixel in data sets that contained images with fire pixels. This result was consistent across all three versions and all three training data set sizes. The control set of 10 non-fire images successfully classified a small proportion of water pixels, with an F1 score of 0.213. Only 0.126 of water pixels were correctly classified as water (i.e., recall). Of all pixels classified as water, 0.702 were correct (i.e., precision), with the remaining being false positives. Successful classification was based heavily on the normalized (0–1) pixel value, with an importance weight of 48.76%, compared to 31.92% and 0.19% for the max-normalized and original values, respectively.

### 3.4. Textural Random Forest Classifier Results

#### 3.4.1. Feature Selection and Importance

Normalization and max-normalization contributed significant information gain both as features themselves and as input parameters to texture and context features. Max-normalization was the most effective technique, and its impact peaked when the imposed maximum was around 1.2 times larger than the minimum non-zero value. Imposed maximums with factors of 1.15 and 1.25 were similarly impactful, with a steep drop-off in performance if the maximum became smaller and modest but steady decreases in performance if the maximum factor became larger. Testing individual texture and context measures on original pixel (i.e., brightness), normalized, and max-normalized values demonstrated that only measures using normalized and max-normalized were meaningful. All measures that were based on original values were discarded for falling below the cut-off F1 score, with most of these features having a feature significance rating of zero. The sole exception was BV, which performed well with the original IR values as input (provided they were median filtered prior to processing).

Across all texture and context measures, disk-shaped kernels tended to perform better than square-based kernels. Entropy and variance filters had the strongest performance, with almost all candidate features being included. Subtracted mean and threshold were very weak features, with all variants being eliminated due to low F1 scores. The six GLCM-based texture measurements considered delivered high information gain, with energy offering the strongest metrics (F1 score of 0.94) and correlation the weakest (F1 score 0.88). Despite the significant information gains available from these measures, they were by far the slowest features to calculate. Each GLCM feature had a computation time that was exponentially larger than all non-GLCM features combined.

### 3.4.2. Classifier Results

After removing all feature variants that produced low F1 scores, a total of 1085 candidate features remained to be evaluated. The initial classifier identified 87 features as contributing no information, all of which were variance-based features combined with a minimum filter. It achieved an overall balanced accuracy of 0.996, with an F1 score of 0.994 and precision and recall scores of 0.995 and 0.993, respectively. Subsequent iterations that discarded low-value features maintained this level of accuracy until 335 features remained. The 91-feature mark was selected as the optimal point, with a balanced accuracy of 0.996 and an F1 score of 0.993, based on the validation data reserved during training. After this point, drops in all scores became more significant with each iteration.

An optimized feature list was derived from the 91-feature version, which eliminated highly correlated features. This reduced the number of features to 21 with negligible drops in accuracy. The optimized version achieved a balanced accuracy of 0.996 and an F1 score of 0.993 based on the reserved validation data. Both the 91- and 21-feature sets were then trained on each image subset (20, 35, and 50 images) for each identified range of trees. Results consistently demonstrated that using large numbers of trees was unnecessary. Accuracy peaked at around 70 trees for all variations, after which additional trees had insignificant impacts in terms of improving accuracy. Classifiers with more training data peaked with slightly fewer trees than those with less.

Based on these results, the 70-tree version of each classifier was tested on all remaining validation images. Both versions achieved high balanced accuracy, F1 score, precision, and recall with additional training data. Validated against all remaining reference images, the 91-feature version attained an overall balanced accuracy of 0.986, with individual images ranging from 0.601 to 0.999. The 21-feature version attained an overall balanced accuracy of 0.984, with individual images ranging from 0.568 to 0.999. The difference in accuracy between the smaller data set (20 IR images) and the larger (50 IR images) was more pronounced in the 21-feature version. In addition, the 21-feature versions with more data typically matched or exceeded the 91-feature versions with less. All variants of the RF classifier exceeded the performance of all other methods (e.g., unsupervised, static GIS) tested at every metric. These results are summarized in Figure 5 and Table 3.

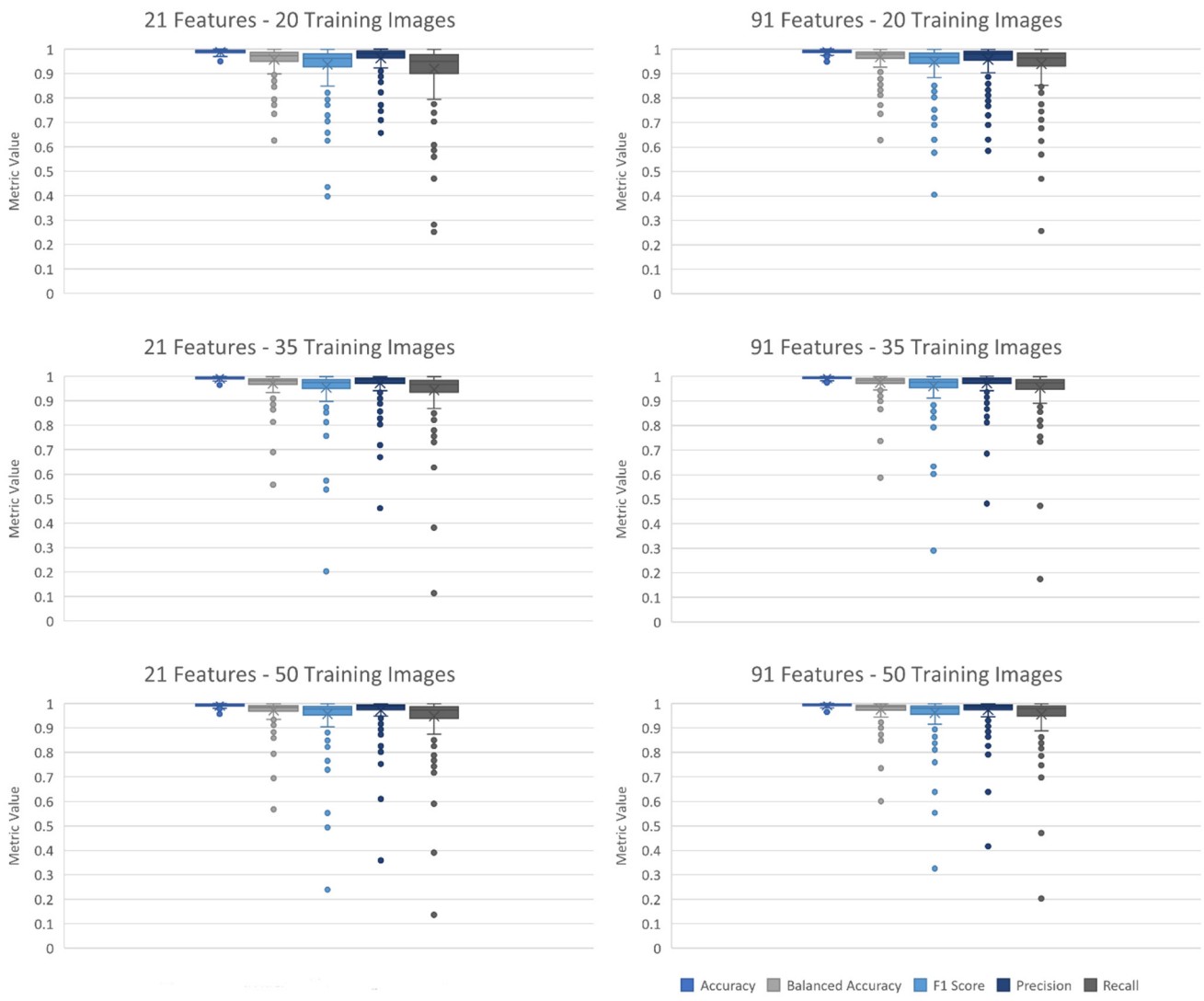

**Figure 5.** Metrics for texture-feature trained random forest classifiers by feature number and training set size.

## 4. Discussion

The static GIS data (CanVec) demonstrate that the use of existing data is functional for obtaining a general idea of where water may be located, but it is not reliable for sensitive mapping. Although the CanVec data set showed an overall strong balanced accuracy (0.965), it consistently marked the boundaries of waterbodies incorrectly (Figure 6). Visual inspection showed that waterbodies typically had oversimplified, jagged edges that were directionally skewed from the edges represented in the aerial IR imagery. This is not surprising given the scale of the CanVec data; however, this is a critical issue for any application that requires accurate delineation. In the context of wildfire mapping, if the CanVec data set were to be used for water segmentation and masking, any fire that burned to the edges of waterbodies would have a reasonable probability of being screened out. It is also worth noting that CanVec is an exclusively Canadian product; similar static GIS data may not exist or be of inconsistent quality depending on the location where mapping is to occur. Although in certain locations these data sets may be of sufficient quality, there are none that currently exist that can be reliably employed across the full Canadian landmass. Accordingly, static GIS data are not recommended for these types of tasks despite the relatively strong performance metrics.

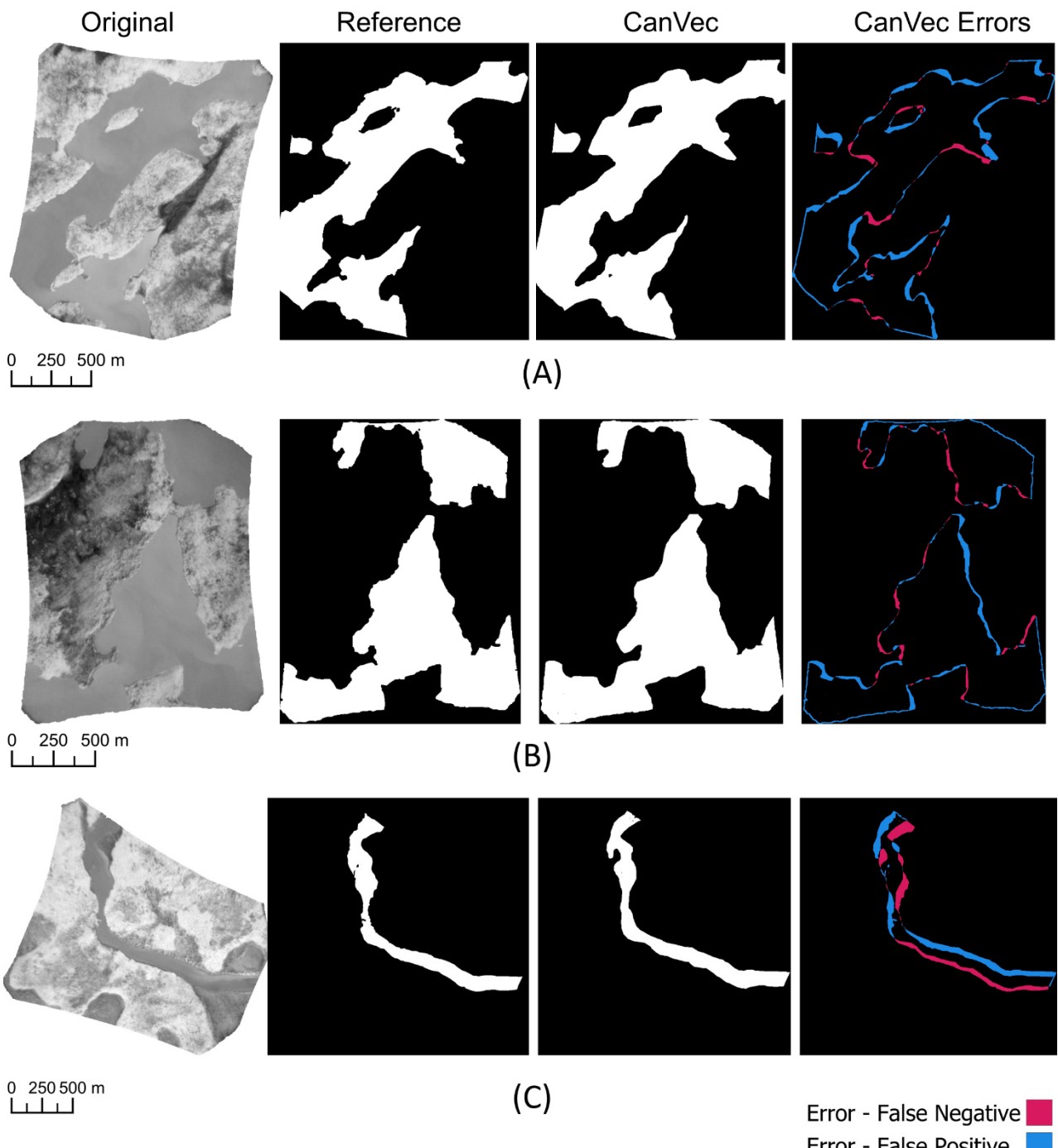

**Figure 6.** Typical CanVec segmentation errors. (**A**) Oversimplification of waterbody edges; (**B**) Directional skewing along shorelines; (**C**) Oversimplification and directional skewing errors typically intensify along narrow waterbodies.

The two unsupervised methods tested offered highly efficient and convenient approaches to waterbody segmentation but were generally not accurate enough for direct applications. The binary entropy (BE) function attained high levels of balanced accuracy. Unfortunately, its performance was inconsistent, with more low-scoring outliers. The binary variance (BV) results were similar, although it was less accurate overall than BE. BE tended toward more accurate waterbody edges, but at the cost of increased false positives that at times occur within areas of active fire (Figure 7A). BV occasionally included thin lines of false negatives in the water, which may coincide with changes in water depth

and therefore surface temperature (Figure 7B). Despite the reduced accuracy, BV was not observed to misclassify fire as water (Figure 7A). Neither method could accurately detect very narrow waterbodies, and both were prone to patches of false positives (Figure 7C). While these methods may have some potential for application in areas where there is neither high-quality static GIS data nor trained classifiers available, they are most valuable as inputs to RF classification, where both methods consistently ranked among the most significant features for classification.

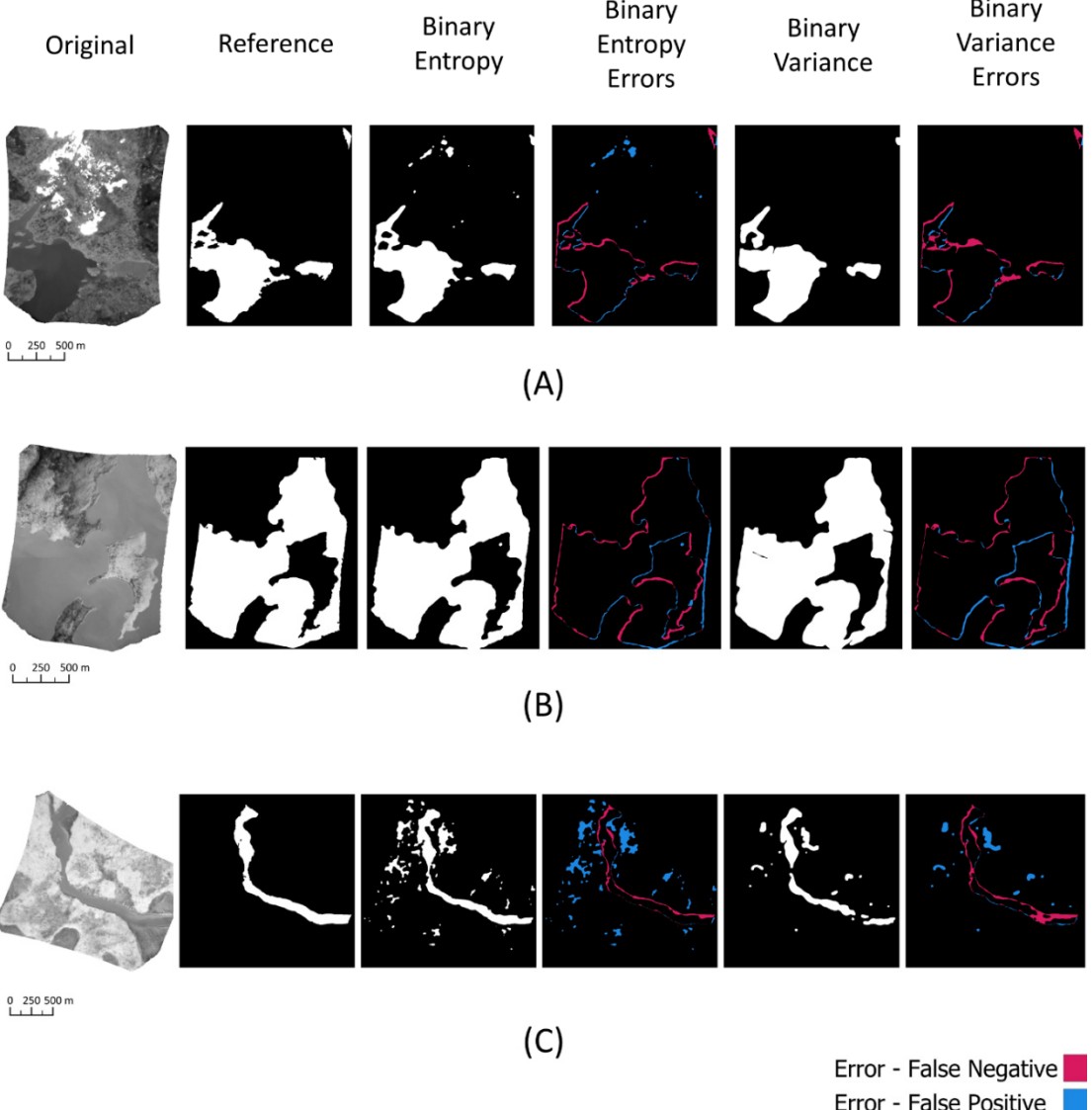

**Figure 7.** Sample segmentation for binary entropy and binary variance (**A**) BE more accurately captures shorelines at the expense of additional false positives, including within areas of active fire; (**B**) BV includes lines of false positives where there are visible temperature shifts in water; (**C**) Both methods are prone to false positive patches, particularly common near narrow waterbodies.

The errors observed for both BE and BV display a consistent pattern of identifying false positives along southern and eastern shores, while false negatives are identified along northern and western shores (Figure 7). Temperature and diurnal trends may be relevant in explaining these errors. As unsupervised methods for segmentation, BE and BV will primarily base their segmentation on the degree of entropy or variance, respectively, within disk-shaped regions. Their success is hinged on water typically displaying much lower levels of entropy or variance than the rest of the landscape, with the level of variance or

entropy sharply rising at the edge of the waterbody. Accordingly, anytime water near the shoreline displays higher levels of entropy or variance in temperature, BE and BV will be prone to displaying false negatives. This also may explain the lines of false negatives that appear to be present in some waterbodies where significant changes in depth may occur (Figure 7B). Similarly, anytime shoreline temperatures are very similar to that of the water, or, the water is at a highly consistent temperature right up to the edge of the shoreline, false positives will occur. These misclassifications reflect a fundamental weakness of unsupervised methods (they cannot learn to recognize other patterns) and suggest that supervised learning approaches may provide a stronger solution.

The baseline RF classifier results confirmed that even if thermal imagery is calibrated, brightness temperature alone is not sufficient information to classify or segment water pixels. Additional context is necessary. Although there was limited success in images that did not contain fire pixels, it did not achieve adequate precision or recall to be valuable.

The feature-based RF classifiers consistently showed combination features as the most significant. Notably, BE, BV, and minimum shifted entropy ($SE_{min}$) ranked among the top 10 features, even across the very early iterations that contained hundreds of candidate features. This reflects the fact that these filters are distilling larger amounts of information (e.g., shifted entropy gathers the most significant value from 5 variations) into a single value. Although clearly beneficial to accurate classification, caution is needed when using these features as inputs due to their significant computational expense (~0.5–1.5 s per feature) when compared to more basic features that can be generated in fractions of a second. Other significant features included scaled entropy stacks ($SE_s$), entropy, mean, and sum (Figures 8 and 9). These features were typically made more significant by processing them a second time through a minimum or maximum filter. This follows the pattern seen with other combination filters, where increased processing appears to concentrate the pixel features into the most significant information for their respective neighbourhoods.

These classifiers had strong, consistent performance across different terrain and on images containing varying levels of fire intensity, including those with fire directly adjacent to the water bodies (e.g., Figure 10A). They did not suffer from the directional skewing found in static GIS data and unsupervised BE and BV segmentation, typically showing only thin scattered strips of pixels along the edges of waterbodies and occasionally at the no-data edge boundary of images (Figure 10A). Some of these misclassified pixels may be due to ambiguities in waterbody boundaries encountered during manual delineation. Many of the classified images display at least small amounts of scattered noise, with a significant number of them capturing larger false positive blobs in areas of uncommonly consistent land surface brightness temperatures (Figure 10B). These larger false positives are a concern, as some are large enough to plausibly be mistaken for a small waterbody. Additional training data may suffice to help reduce these larger errors. Adding a post-processing step, such as a majority filter or eroding edges of the waterbodies, could help clean up the smaller scattered false positives and improve accuracy, but care would need to be taken to ensure that it does not eliminate true islands or interfere with shoreline accuracy. Finally, as with all other methods explored in this study, accuracy was weaker when there was a smaller proportion of water pixels in the frame or when waterbodies were narrow (Figure 10C). Narrow waterbodies or streams are an inherent weakness in this style of segmentation because most features consider the consistency of the pixel brightness temperature values across a given neighbourhood. Narrower sections of water do not demonstrate the same level of neighbourhood consistency, making them difficult to detect.

Further work may be undertaken to improve the efficiency of this method. Given how well the 21-feature variant of the classifier performed in comparison to the 91-feature version, it is assumed that there is room for further feature optimization that would improve upon efficiency and reduce complexity. Additionally, some of the features that were discarded as insignificant for falling below the F1 score threshold in the original round of selection may still provide additional context that could benefit classification and, therefore, segmentation. While the finished classifiers presented here perform well, there is computa-

tional expense to generate each of the features necessary for classification. To further reduce feature generation expense, parallel programming should be explored to improve efficiency. This study used parallel processing exclusively for training and classifying images but not for generating features or for simultaneous processing of multiple images.

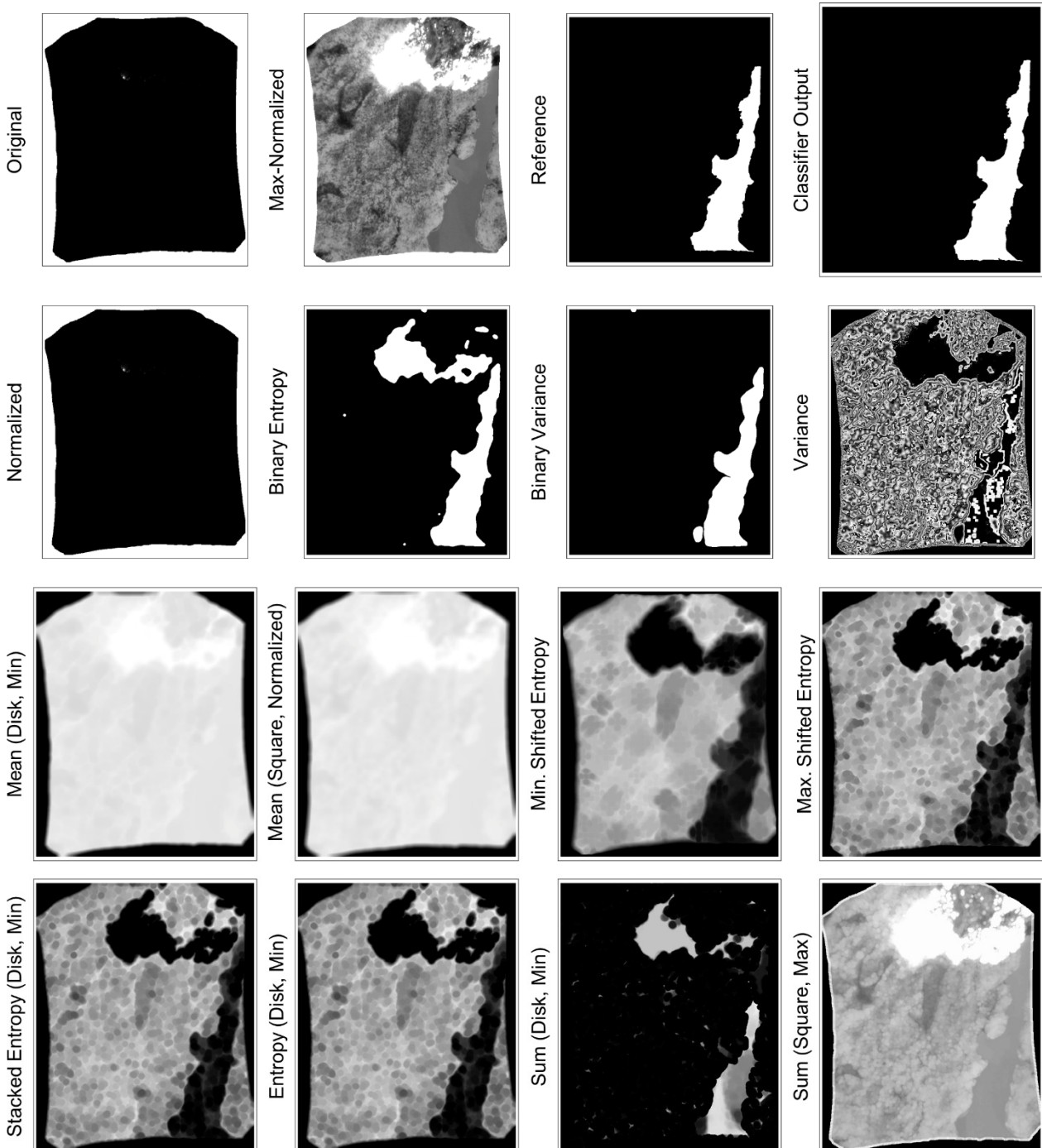

**Figure 8.** Sample of significant features for a fire pixel-containing IR image. Original and normalized images reflect the suppressed detail that can occur when IR images containing active wildfire are processed by standard image processing libraries. As described in Appendix A, the max-normalization technique can help preserve this detail.

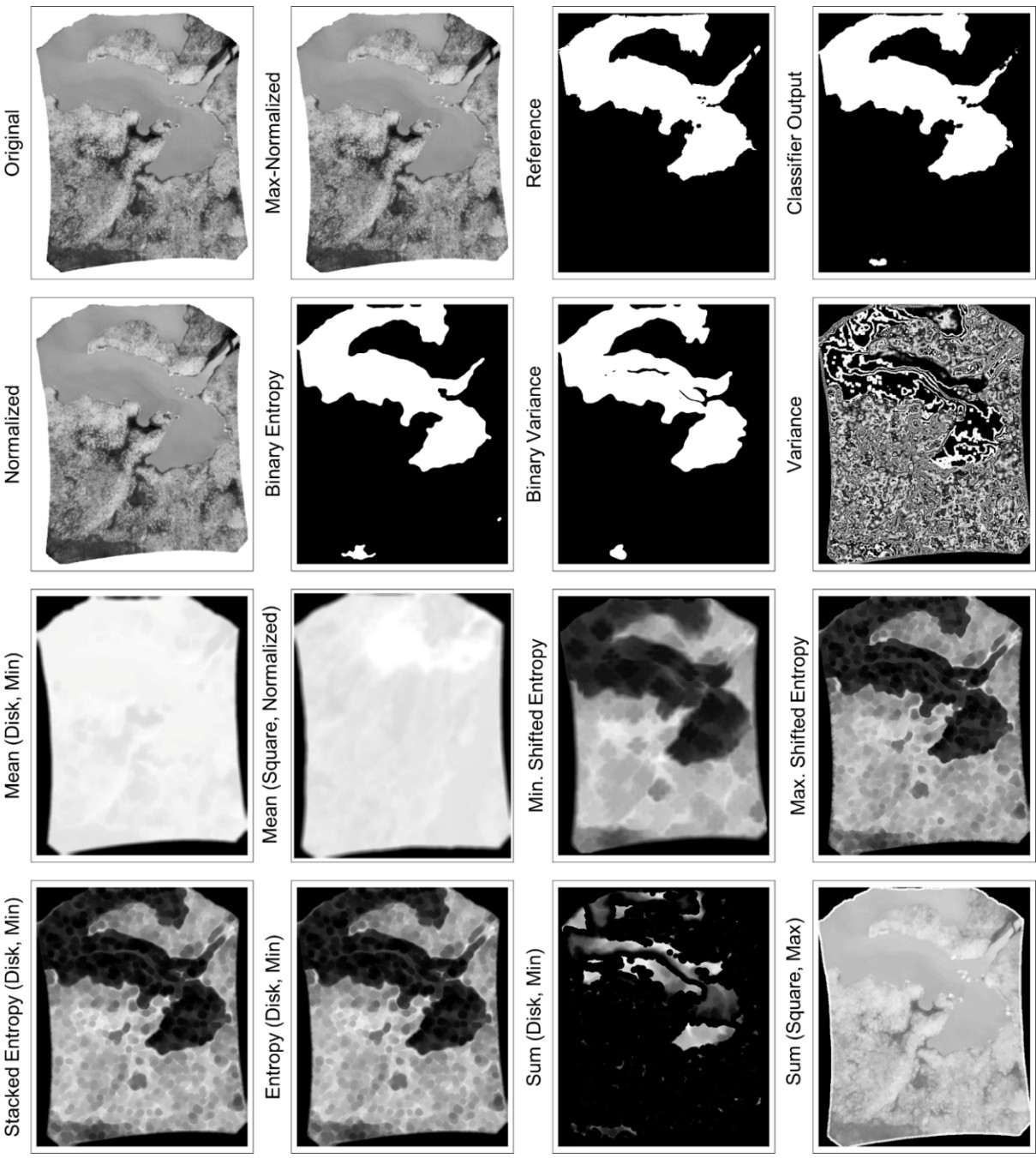

**Figure 9.** Sample of significant features for IR image without fire pixels.

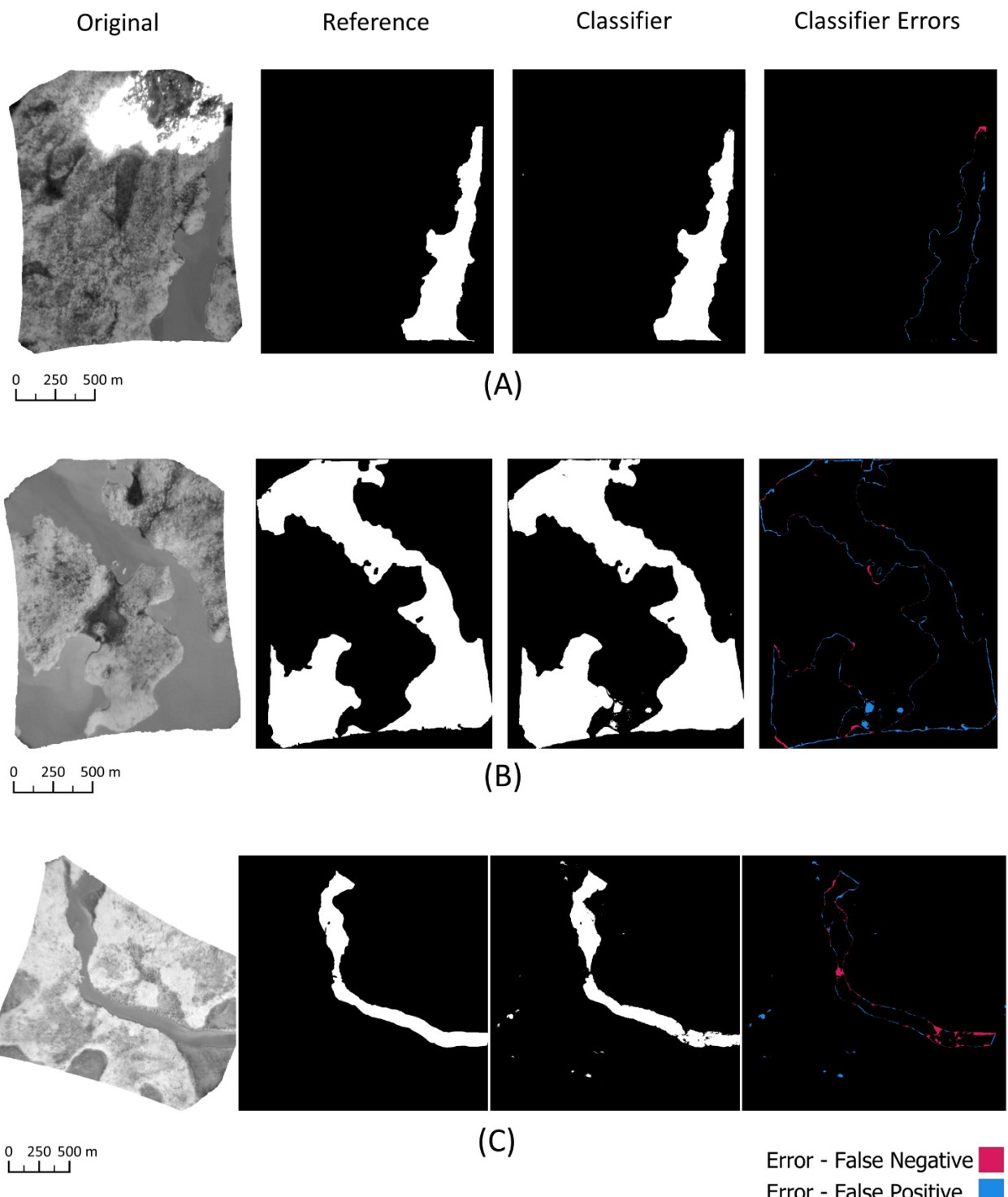

**Figure 10.** Sample segmentation results using the 91-feature RF classifier trained with 50 images. (**A**) Overall strong accuracy, even in the presence of flaming combustion close to the shoreline, without the directional skewing common in static GIS data output; (**B**) noise and larger false positive blobs occurred in many images; (**C**) weaker performance near narrower waterbodies.

More efficient approaches may also make some of the excluded features viable. Although GLCM texture measures ultimately were discarded in the final classifiers used in this study, they remained significant until very late in the iterations of feature selection. They may be even more valuable in other contexts, where features of interest have less

homogeneous texture than water pixels. These measures have the potential to contribute useful and unique information but were found to be very cumbersome due to their extreme computational expense. The inefficiency of these calculations may have been in part due to the language and implementation used. Further consideration, with a closer look at parallelization and optimization, is needed to fully evaluate their usefulness and feasibility for time-sensitive tasks in IR images.

The inherent challenge in detecting small and narrow waterbodies is shared across the RF, BE, and BV classifiers. When examining the performance of each method relative to the target waterbody size, all three methods performed poorly for waterbodies less than 1 ha (Figure 11). That said, it is notable that the performance sharply improves above 1 ha for the RF (Figure 11A), while the other methods improve more gradually with increasing waterbody area (Figure 11B,C). Accordingly, it is clear in this comparison that the RF approach is the superior method tested here.

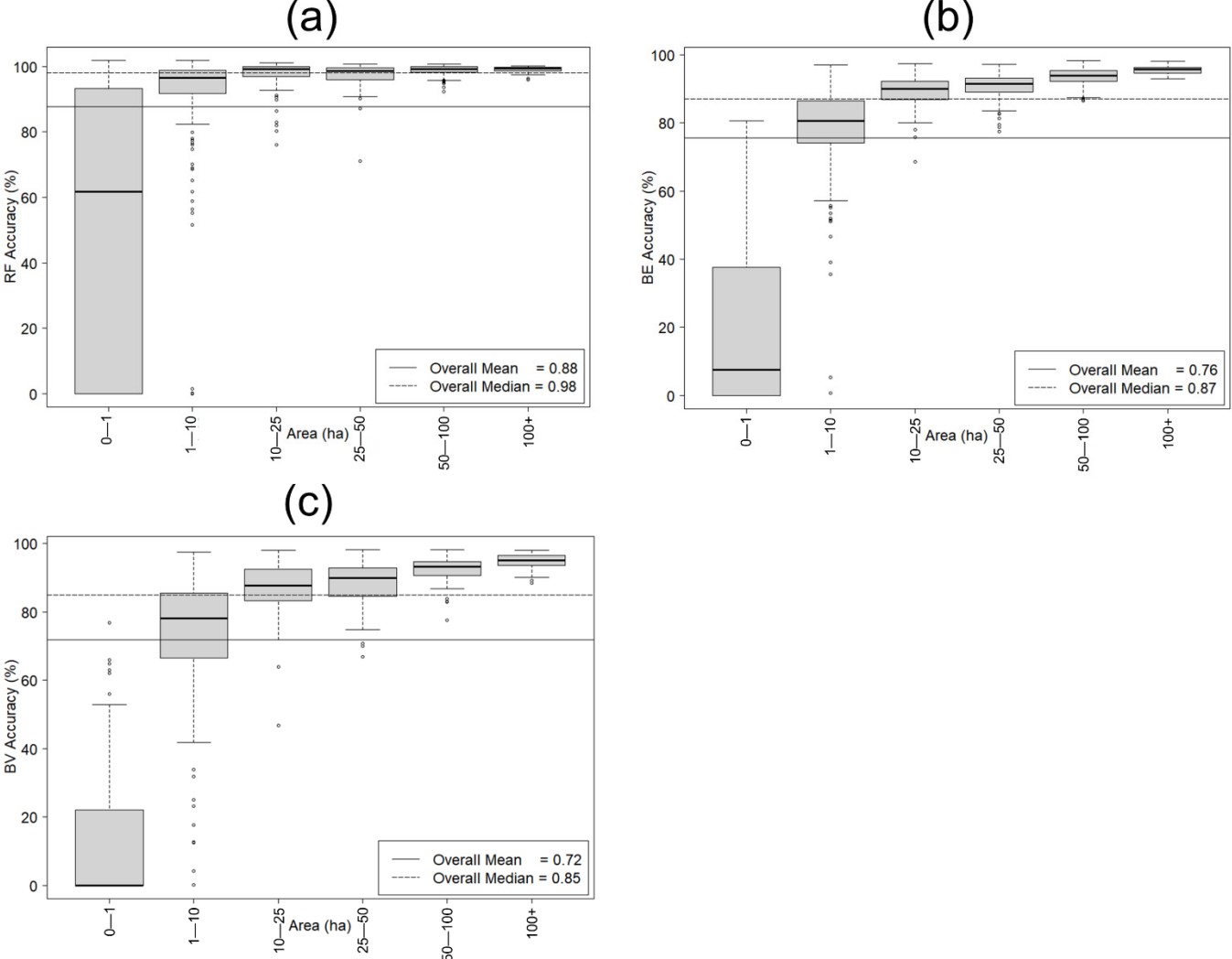

**Figure 11.** Comparative distribution of classifier accuracy with respect to waterbody area for all 517 waterbodies binned by size classes for (**a**) RF, (**b**) BE, and (**c**) BV.

One potential limitation of this study was the use of a single interpreter to perform manual delineation of waterbody boundaries. Although this approach is consistent with previous studies examining IR imagery for RF mapping applications (e.g., [44]), the level of bias and the degree to which this impacted the overall accuracy assessment of each method could not be measured.

## 5. Conclusions

This study focused on identifying a technique that can successfully segment waterbodies in IR imagery containing heterogenous wildfire activity, with a particular focus on identifying what features would make such segmentation possible for an RF classifier. Promising performance was seen in binary entropy and binary variance approaches (median accuracies of ~87% and ~85%, respectively). However, the results confirmed that while isolated thermal IR pixel values alone cannot be used to segment waterbodies, RF can be highly successful (median accuracy ~98%) when texture-based features are included as model inputs, especially when those features are based on entropy and variance. The outputs of this classifier achieved waterbody delineation that was more accurate, particularly along shorelines, than seen in static GIS data. Normalizing and imposing maximum values on data were essential steps for successful classification and for mitigating the interference caused by intense wildfire pixels. Variations of entropy calculations that considered entropy across different neighbourhoods, as well as the minimum and maximum values within those regions, were of particular importance for accurate classification. The unsupervised methods considered did not perform well independently but provided valuable inputs to the RF classifier. Notably, the binary entropy method was highly efficient and delivered levels of accuracy that were comparable (though less consistent) to static GIS data.

Although processing time needs to be considered, the high accuracy of this method makes it a strong candidate for operational use. In addition to providing this step toward the automation of wildfire perimeter mapping, the excellent performance achieved in this study suggests a possible mechanism for developing highly accurate labelled training data sets. Currently, many deep learning methods (in particular, convolutional methods that would likely improve results) for waterbody segmentation and thermal imagery are not viable due to requirements for large volumes of training data. However, the present segmentation technique offers a possible method for expediting the creation of reference data, making these more complex techniques feasible.

**Author Contributions:** Conceptualization, J.A.O., F.C.P., Q.T., A.S.C. and J.M.J.; methodology, J.A.O.; software, J.A.O.; validation, J.A.O.; formal analysis, J.A.O.; resources, J.M.J., A.S.C. and M.J.W.; data curation, J.A.O. and A.S.C.; writing—original draft preparation, J.A.O.; writing—review and editing, all authors; supervision, F.C.P., Q.T., A.S.C. and J.M.J.; project administration, F.C.P., Q.T. and J.M.J.; funding acquisition, J.M.J. and M.J.W. All authors have read and agreed to the published version of the manuscript.

**Funding:** Aspects of the field campaign providing the data for this study were part of the Fire Detection Experiment (FIDEX) conducted under a program of, and funded by, the European Space Agency (ESA Contract No. 4000122813/17/I-BG), from NERC National Capability funding to the National Centre for Earth Observation (NE/Ro16518/1) and Leverhulme Trust grant number RC-2018-023 to the Leverhulme Centre for Wildfires, Environment and Society. Aspects of this study were supported through a Collaborative Research Agreement between AFFES and the Canadian Forest Service (CFS), as well as the Canadian Safety and Security Program Project Charter CSSP-2019-TI-2442 between CFS and Defence Research and Development Canada.

**Data Availability Statement:** The data used to support the findings of this study are available from the corresponding author upon request.

**Acknowledgments:** The NERC Airborne Research and Survey (Airborne Remote Sensing Facility) and British Antarctic Survey are thanked for their support of the flight campaign, without which this research would not have been possible. The authors would like to thank the Ontario government's Aviation Forest Fire and Emergency Services (AFFES) division for their logistical and operational support in executing the field campaign, with special thanks to the Red Lake Fire Management Headquarters and Colin McFayden. We also thank Alexander Charland for providing preliminary data management support.

**Conflicts of Interest:** The authors declare no conflict of interest.

## Appendix A

*Appendix A.1. Feature Selection and Importance*

Appendix A.1.1. Normalization and Maximum Values

Although often considered unnecessary for tree-based classifiers such as RF, normalization was evaluated as a potential feature. The precise values contained in IR imagery are unit- and calibration-dependent but, in all cases, are unlikely to conform to the range expected by existing image processing and filtering libraries (which generally anticipate values from 0 to 1 or 0 to 255). Many library functions will automatically convert values into the expected range, but such conversion frequently truncates the resulting values, leading to a substantial loss of detail for very small and very large values. For this reason, all texture measures/techniques were evaluated with both original values and pre-emptively normalized values.

In the presence of fire pixels, normalizing inputs can suppress detail. With a fire pixel potentially being several hundred times more intense than non-fire pixels, such scaling can result in the most intense pixels dominating the scale and all other pixel values (including less intense fire pixels) becoming negligibly close to zero. To address this, normalization with an imposed maximum value (max-normalization hereafter) was also evaluated. This concept was based on the observation that the minimum non-zero pixel value had low variance across all images, despite the extreme variance in maximum values. It was implemented by defining an artificial maximum as some multiple of the minimum non-zero value in the image and then setting any values that exceed that value to be equal to it. Potential factors ranging from 1.1 to 2.5 times the minimum non-zero value were evaluated in increments of 0.05. The factor identified as providing the most information retention was used as a feature input to evaluate texture and context measures.

Due to the discovered inability of RF to classify water pixels based on temperature alone, the max-normalization candidates had to be evaluated in conjunction with a textural or contextual filter to make their impacts visible. To address this, the tests were repeated twice: once using an entropy feature and once using a neighbourhood mean. Repeating across two very different features was intended to ensure that the max-normalization performance trend was consistent across features rather than specific to a particular feature.

Appendix A.1.2. Texture and Context Measures

The initial measures of texture and context considered were local entropy, mean, subtracted mean (i.e., difference between pixel value and mean for its local neighbourhood), sum, threshold, and variance (based on the sum of squares). Entropy and variance were also tested as features applied to the results of all other features, including one another (e.g., entropy was evaluated on the results of the variance filter). Each resulting feature was also combined with a minimum and a maximum filter to generate additional variations.

Haralick et al. [30] introduced 14 textural feature measurements for image processing; these metrics primarily focus on calculations surrounding a grey-level co-occurrence matrix (GLCM). Six of these texture measurements were considered here as additional features: angular second moment (ASM), contrast, correlation, dissimilarity, energy, and homogeneity.

Appendix A.1.3. Filter Combination Features

Five unique filter combinations were also created to be evaluated as feature inputs: scaled entropy stack, minimum shifted entropy, maximum shifted entropy, binary variance, and binary low entropy.

The scaled entropy stack ($SE_S$) was developed in response to the impacts of fire pixels in images. When fire pixels are present, the extreme variation in temperature among them results in all but the most intense fire areas receiving an entropy value of near-zero. Although the max-normalization technique discussed above addresses this to some extent, it results in large areas of very low entropy wherever fire pixels do occur. The $SE_S$ addresses this shortcoming by first calculating the entropy for the entire image with the maximum

value in the image capped at double the minimum non-zero value. The entropy is then calculated for the entire image again, this time with a maximum value that is double that used in the previous iteration. The two entropy image matrices are then merged together, with each pixel assuming the value of whichever matrix assigned it the largest entropy level. This process is repeated until the imposed maximum value for the purpose of entropy calculations is equal to or exceeds the actual maximum value in the original image.

Minimum shifted entropy ($SE_{min}$) and maximum shifted entropy ($SE_{max}$) are each based on calculating local entropy from five different positions within a local neighbourhood: in the centre of the kernel and from the centred leftmost, rightmost, top, and bottom positions. The five potential pixel values are then compared, with only the minimum or maximum being retained for $SE_{min}$ and $SE_{max}$, respectively. The intention was to capture the most significant entropy value from the pixel's neighbourhood without the increased dimensionality of including each positional measure as a separate feature.

The binary variance (BV) feature was inspired by the rule-based approach to waterbody segmentation that was introduced by Wilson [41]. In his approach, Wilson identified water seed pixels based on band values and a $5 \times 5$ variance filter value of 0 or 1. From these seeds, water regions were grown. In the BV feature here, variance is calculated using a $2 \times 2$ sliding window. In the resulting variance matrix, regions with a value of 0 are treated as water, and all non-zero areas are treated as not water. The water regions are expanded using several iterations of morphological dilation and then contracted using an equal number of iterations of morphological erosion. Finally, a series of majority filters are passed over the binary image to smooth the edges and reduce noise.

The binary low entropy (BE) feature follows a similar procedure to BV. An entropy filter is first passed over the input image. If the input image has a maximum value larger than double its non-zero minimum value, a second entropy filter is applied, with an imposed maximum value of double the non-zero minimum. The results of the two entropy filters are combined, with the largest value from either matrix retained. The entropy matrix is then normalized and passed through a median filter. The median result is then converted to a binary image with the criteria that entropy less than 0.3 is water and entropy values greater than that are non-water. A minimum filter is passed over the resulting binary matrix several times to reduce the noise caused by small clusters of false positives. Finally, a maximum filter followed by a morphological close filter is reused to regrow lost regions from the minimum filters and close any small gaps in large waterbodies.

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
