# Peer review of "A Machine Learning Approach to Waterbody Segmentation in Thermal Infrared Imagery in Support of Tactical Wildfire Mapping"

_remotesensing, doi:10.3390/rs14092262_

Round 1
Reviewer 1 Report
Overall, a weary interesting paper that lacks some additional analysis of results and performance.
Q1:
In line, 257 authors introduced normalization (that was later explained in the appendix). Was used sensor radiometric calibrated (where body temperature can be directly calculated from thermogram). Was ambient temperature taken on to the account?
In appendix, the authors mentioned that fire-related pixels are brighter the rest of the pixels. Can some additional pre-processing of thermograms besides normalization be done, like histogram equalization, or even CLAHE?
Q2 many reference errors, like Line 271, 274,
The images filters considered are listed in Error! Reference source not found..
Q3:
Please provide the configuration which executed algorithms
Running at an average of 0.519 seconds per image on a machine based on
Q4:
Comment, is there any wildfire present near water body sections? (something is commented in line 500) Are they classified correctly? (subjective analysis)
Q5:
What is the minimum waterbody size considered?
Q6:
Can edges of water bodies be eroded (in both labeled images and extracted images, thus omitting some small errors and uncertainties around water body edges?)
Q7:
Paper lacks some basic analysis considering water body sizes on images, and also fire contaminated areas on the image (in terms of pixels)
Are they usually connected to a larger area?, or scattered over the image.
Q8:
Accuracy/ precision was calculated on a pixel level, as the authors stated „Overall precision score (i.e., percentage of pixels classified as water that are accurately 381 classified)“
It is crucial to provide an analysis of how many water body objects in the image are correctly calculated, and their size distributions, as the ratio of body objects can be unbalanced on some image segments.
Q9:
Considering the problem of classifying Narrower sections of water (line 586), 2D image convolution used in common ML algorithms may detect that water bodies better if given enough training data…
Is the convolution-based approach even considered?
Reviewer 2 Report
This is a very interesting study that explores the use of the Random Forest (RF) classifier for the segmentation of water bodies in thermal IR images containing heterogeneous forest fires. The results of the classifier are compared with static data as well as with the results of two unsupervised segmentation techniques based on entropy and variance.
Very promising results are shown with very balanced and very high accuracies (> 98.6%) for thermal images with and without wildfire pixels, with an overall F1 score of 0.98.
Some minor areas of improvement are discussed below:
The second figure has been mistakenly named as figure 1 and thereafter all other figures have the incorrect numbering. Revise therefore also the references to the figures once renumbered.
In line 271, 274, 285, 313, 328, 331, 387, 409, 418, 486 there are errors of reference not found.
I think that in Table 2 we should be consistent with the different figures of each metric and put not only the mean values, but also the variance.
I also think it would be interesting to mark in bold the maximum values per row, to better see which method is better in each metric.
Figures 7 and 8 do not follow the same ordering of the images as in the rest and the truth is not easy to follow. I think it is clearer to show first the original, then the reference and then the different results, as in figure 6 for example.
Are the original and normalized image in figure 7 correct? They do not seem to be, since they do not resemble at all with the reference, which is also in the opposite corner.
In the conclusions an express mention of the results obtained and commented in the discussion is missing.
Reviewer 3 Report
This paper proposes an assessment of the effectiveness of existing waterbody segmentation techniques in thermal IR imagery collected during wildfires. The authors have conducted extensive literature review of previous studies in this domain. The manuscript deals with an interesting topic and would help wildfire management agencies. Nevertheless, I have some comments
Lines 36-147: In introduction; please provide a short introduction of your proposed methodology adopted to achieve the aim of this study (lines 141-147). Such introduction (available in lines 149-160) should follow line 147. Please relocate lines 149-160 in the introduction.
Line 148: The section, 2. Materials and Methods, should be restructured: please start this section with the description of the area of study (2.1), Followed by methodology (2.2). In this section (2.2), please provide the description of the dataset Airborne Imagery, Ancillary Data and Training and Validation Reference Data. Please provide more details of how Training and Validation data were sampled.
Line 174: What type of accuracy are you referring to?
Lines 174-187: Please indicate if you analyzed individual images or if you mosaiced these 2,496 subsets into one orthophoto which was further analyzed.
Lines 372-373: please provide more details or equations on how these metrics accuracy were calculated
